# Organisation and characteristics of out-of-hours primary care during a COVID-19 outbreak: A real-time observational study

**Stefan Morreel**  *, **Hilde Philips, Veronique Verhoeven**

Department of ELIZA (Primary and Interdisciplinary Care), University of Antwerp, Antwerpen, Belgium

* stefan.morreel@uantwerpen.be

**Data Availability Statement:** Data availability statement: Given the privacy policy of the iCAREdata database, the authors are not allowed to

## Abstract

### Background

During the COVID-19 pandemic, general practitioners worldwide re-organise care in very different ways because of the lack of evidence-based protocols.

### Objective

This paper describes the organisation and the characteristics of consultations in Belgian out-of-hours primary care during five weekends at the peak of a COVID-19 outbreak and compares it to a similar period in 2019.

### Methods

Real-time observational study using pseudonymised routine clinical data extracted out of reports from home visits, telephone- and physical consultations (iCAREdata). Nine general practice cooperatives (GPCs) participated covering a population of 1 513 523.

### Results

All GPCs rapidly re-organised care in order to handle the outbreak and provide a safe working environment. The average consultation rate was 222 per 100 000 citizens per weekend. These consultations were handled by telephone alone in 40% (N = 6293). A diagnosis at risk of COVID-19 was registered in 6692 (43%) consultations,. Out of 5311 physical consultations, 1460 were at risk of COVID-19 of which 443 (30%) did not receive prior telephone consultation to estimate this risk. Compared to 2019, the workload initially increased due to telephone consultations but afterwards declined drastically. The physical consultation rate declined by 45% with a marked decline in diagnoses unrelated to COVID-19.

### Conclusions

General practitioners can rapidly re-organise out-of-hours care to handle patient flows during a COVID-19 outbreak. Forty percent of the out-of-hours primary care contacts are handled by telephone consultations alone. We recommend to give a telephone consultation to all patients and not to rely on call takers to differentiate between infectious and regular care.

share the used database. Sharing this database would potentially harm the privacy of the included patients as one might get information about their identity by combining data from several columns (variables). We are however able to deliver a selection of columns upon reasonable request. A part of the iCAREdata database is disclosed to the public on a website (https://icare.uantwerpen.be) this includes the data presented in this article but with less detail. We do not have ethical clearance to share our data. As an alternative we have added the output of our statistical software to the supplementary material. Access to our data can be requested by contacting icaredata@uantwerpen.be.

**Funding:** All authors received grant number T000718N from the Research Foundation - Flanders (FWO, see https://www.fwo.be/). The funder had no role in study design, data collection and analysis, decision to publish, or preparation of the manuscript.

**Competing interests:** I have read the journal's policy and the authors of this manuscript have the following competing interests: All authors have completed the ICMJE uniform disclosure form and declare: Stefan Morreel is an unpaid Board member of one of the participating General Practice Cooperatives and is paid by the Belgian ministry of health to coordinate the response of this cooperative to the COVID-19 pandemic. Veronique Verhoeven has participated at the same GPC by handling telephone calls in a fee-for-service model. All authors are board members of iCAREdata as part of their academic position, the database used in this study. This does not alter our adherence to PLOS ONE policies on sharing data and materials.

The demand for physical consultations declined drastically provoking questions about patient's safety for care unrelated to COVID-19.

## Introduction

Novel coronavirus disease 2019 (COVID-19) provoked by SARS-CoV-2 is a spreading threat and its clinical and epidemiological characteristics are still being documented. [1, 2] The current COVID-19 pandemic puts extreme stress on healthcare organisation. In several countries including China, Italy, Brazil, Spain and certain parts of the United-States, the demand for emergency and intensive care exceeded the available resources.

Almost all countries are struggling to tackle this pandemic in different ways including different strategies for primary care. [3] in the UK roughly 75% of patients is seen remotely, in the USA primary care offices are capable of managing patient flows across home, clinic, hospital, and post-acute care [4], Columbia uses a very similar approach with primary care as a gate keeper [5] and finally, Australia has decided to unprecedented level of support for the primary care system. [6]

On 2020/04/20 Belgian health authorities reported a total number of 39 983 infections, 13362 hospitalisations and 5 828 deaths (including 3028 suspected case in homes for the aged). The peak of the current outbreak was situated at the beginning of April, a partial lockdown was initiated on march 13. [7] All patients presenting with symptoms of acute respiratory infection were considered as suspected cases as tests were not available in primary care services. These patients needed to stay at home in self-isolation during at least seven days. [8] The government's recommendation to always call a doctor before going to a practice or ED was omnipresent in the media.

All chronic care both inside and outside hospitals had been suspended and emergency plans had been activated in order to increase the number of beds available in intensive and emergency care. This might have led to delayed access to hospital care and consequently increased morbidity and mortality unrelated to COVID-19. [9, 10]

Almost all Belgians are member of the mandatory healthcare insurance. Emergency Departments (EDs) and General Practitioners (GPs) are freely accessible, they are paid by a fee-for-service system. Patients pay 18% of their healthcare expenditures themselves. [11] Every ED in Belgium needs to give appropriate care to anyone entering the service regardless of citizenship, legal status or ability to pay.

In many European countries, out-of-hours primary care is increasingly organised in large-scale General Practitioners Cooperatives (GPCs). As a bottom-up response these GPCs have adapted to the COVID-19 pandemic. [12] The COVID-19 pandemic has a substantial impact on primary care consultations. [13] In order to prevent infection of patients and healthcare professionals, a shift from in-person to remote consulting by telephone or video is occurring. [14] Before the pandemic, remote consultations were neither reimbursed nor deontologically allowed. Employees in Belgium unable to work due to medical problems need a medical certificate and for this reason have to see a medical doctor. During the COVID-19 pandemic, doctors are allowed to deliver it after a telephone consultation.

During office hours, the organisation of GP-practices in Belgium is very diverse making rapid research difficult. During weekends, the organisation of care is much more uniform due to the existence of GPCs. Using the iCAREdata database containing routine data in out-of-hours care in Belgium, quick analysis of all contacts at selected GPCs is possible. [15, 16] In

this paper we assess the characteristics of GPC consultations. We compare these characteristics to a reference period in 2019 and describe the organisational changes the GPCs made.

## Methods

This study was approved by the Ethics Committee of the Antwerp University Hospital (reference 20/14/170). The boards of all participating GPCs gave consent; individual patient consent was waived because we only used pseudonymised data.

We included 9 GPCs out of the 13 GPCs connected to iCAREdata. They all use the same software (Mediris 2.4®). One GPC was excluded because it is located within an ED which is not the focus of this study; three GPCs were excluded because the quality of data was insufficient. The included GPCs cover an average population of 168 169 citizens each (range 85 870–251 000) and have an average of 192 member GPs (range 100–380). In total they cover a population of 1 513 523 citizens and 1730 GPs (including trainees). All GPs having a practice in a community covered by a GPC are obliged to work approximately one shift a month in that GPC. The studied GPCs are all located in the Dutch speaking part of Belgium (Flanders) and cover 23% of the Flemish population. For the historical reference period in 2019, four additional GPCs were excluded because they did not yet deliver reliable data.

All included GPCs filled in an e-mail questionnaire about their organisation in order to understand the data collected. This questionnaire covered the following subjects: location of possible COVID-19 consultations, number of citizens covered by the GPC, number of active GPs covered by the GPC, starting date of specific COVID-19 care, profile and tasks of call takers, availability of consultations without appointment, organisation of telephone consultations and collaboration with surrounding EDs. The first author called the manager or a board member of all GPCs to verify the collected information.

Clinical data registered during home visits, telephone- and physical consultations were collected during five weekends (Friday 7 pm to Monday 8 am) and one bank holiday (Easter Monday) in 2020 and 2019.

During a telephone consultation, the software allows the GP to electronically prescribe drugs, deliver medical certificates and refer to an ED or the patient's own GP. GPs are obliged to fill in a diagnosis using a Belgian list of clinical labels linked to the International Catalogue of Primary Care (ICPC-2). This list was extended with four labels related to COVID-19 based on SNOMED-CT concepts: confirmed case, suspected case, case with close contact to confirmed case and fear of COVID-19 without clinical suspicion. Because not all GPs were aware of the existence of these labels some have used other diagnostic labels or just a symptom as a diagnosis instead. The authors SM and VV independently rated all labels used in this study as at risk for COVID-19 or not. All labels concerning a diagnosis of respiratory tract infections or a symptom in a list of the most common COVID-19 symptoms (fever, cough, myalgia, fatigue, expectorations, conjunctival congestion, diarrhoea, loss of smell and taste and dyspnoea) were rated as at risk. [17–19]

All data were automatically available in iCAREdata: a research database on out-of-hours primary care. iCAREdata uses the pseudonymised Belgian national number as a unique identifier making it possible to link several consultations (even at different GPCs) to one patient but automatically excluding patients without such a number. [15] It contains most of the data fields from the software used in the GPCs.

Data was analysed using JMP 14®. Pearson chi-square testing was used for all comparisons in categorical variables. A two sample student's t-test was used to compare the means of age between the consultation types and the historical reference group.

## Results

### Organisation of COVID-19 care

See Fig 1 for a schematic overview of the organisation of the GPCs during a COVID-19 out-break. Seven GPCs reorganised their entire care paths immediately after the start of the Belgian

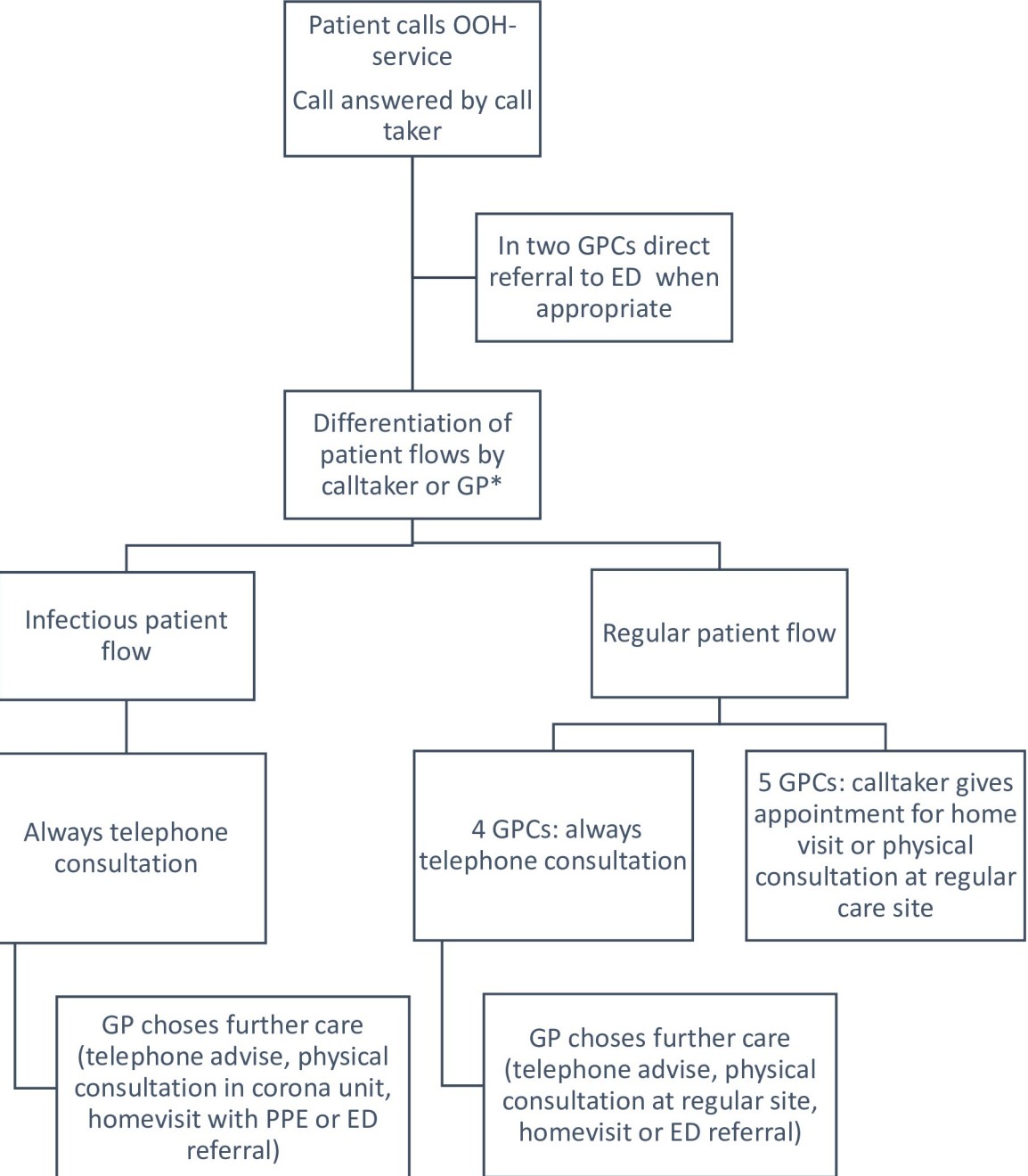

**Fig 1. Organisational model during a COVID-19 outbreak of the nine included General Practice Cooperatives.** *in five included General Practice Cooperatives the call takers make this differentiation. In four included General Practice Cooperatives, the general practitioners make it themselves. In those GPCs, a telephone consultation within the regular patient flow can result in a referral to the infectious patient flow. PPE: Personal Protective Equipment.

lock-down; the other two made this switch one week later at the start of this study. They all provide staff with personal protective equipment and clean rooms for physical examination of patients at risk for COVID-19 called Corona Units, located outside of the sites for regular care. Due to the structure of the iCAREdata database it was not yet possible to make a distinction between the consultations by site.

Patients have to call prior to presentation at the GPC, including those presenting themselves spontaneously at the GPC. A call taker registers the administrative data. In two GPCs, the call takers refer selected patients directly to the ED (using a local triage protocol [20]) while all other GPCs do not perform any kind of triage. In five GPCS, call takers differentiate the patients between an infectious patient flow (suspected COVID-19) and a regular patient flow (no suspicion of COVID-19)) by checking the most common COVID-19 symptoms (fever, dyspnoea, coughing, sneezing, running nose, throat ache and feeling unwell). In the other four GPCs, the GP makes this differentiation based upon personal opinion. Within the infectious patient flow, all GPCs perform a thorough telephone consultation, when needed the GP will refer patients to a home visit with PPE, a physical consultation at the corona unit or referral to the ED. Within the regular patient flow, four GPCs perform a telephone consultation on all patients after which the GP decides upon the following care. In the other five GPCs, the call taker gives an appointment for home visit or physical consultation at a regular care site.

## Characteristics of GP consultations

We included 15655 consultations by 12096 unique patients during five weekends in 2020, 8942 (57%) of these were telephone consultations of which 6293 (70%) could be handled by telephone alone. The 2019 reference group contains 7571 physical consultations.

See Table 1 for characteristics of the consultations. In 2020, for 6692 (43%) consultations a diagnosis at risk of COVID-19 was registered. This rate declined from 2055 (58%) in the first weekend to 533 (27%) in the final weekend. The most common diagnoses at risk of COVID-19 were: suspected case of COVID-19 (N = 3131, 47%), unspecified viral infection (N = 618, 9%), fever (N = 602, 9%), acute upper respiratory tract infection (N = 546, 8%) and coughing (N = 417, 6%). During 1438 (9%) consultations a medical certificate for employees was delivered. This proportion was higher in the first weekend (N = 422, 12%) and lower in the final weekend (N = 153, 8%, p<0,001). The GPs referred 868 patients (6%) to the ED. For patients with a diagnosis at risk of COVID-19, the referral rate to the ED for telephone consultations was lower (N = 96, 2%) whereas it was higher during physical consultations (N = 141, 10%,

**Table 1. Characteristics of included consultations\*.**

| | All consultations reference period 2019 (n = 7571) | All consultations 2020 (n = 15655) | Telephone consultations (n = 8924) | Physical consultations (n = 5311) | Home visits (n = 1420) |
|---|---|---|---|---|---|
| Female | 4102 (54%) | 8671 (55%) | 4894 (55%) | 2921 (55%) | 856 (60%) |
| Age (years): mean | Mean: 38 | Mean: 43 | Mean:41 | Mean:39 | Mean:74 |
| Age (years): SD | 28 | 25 | 24 | 22 | 21 |
| Suspected case COVID-19 | N/A | 6692 (43%) | 4894 (55%) | 1460 (27%) | 338 (24%) |
| Medical certificate for employees delivered | 1200 (16%)* | 1438 (9%) | 759 (9%) | 662 (12%) | 17 (1%)* |
| Follow-up by own GP | 365 (5%)* | 969 (6%) | 652 (7%) | 260 (5%) | 57 (4%)* |
| Referral to ED | 481 (6%)* | 868 (6%) | 212 (2%) | 496 (9%) | 160 (11%)* |

SD: Standard Deviation, N/A: not applicable

*: all p-values < = 0,001

**Table 2. Number of consultations per 100 000 citizens during a COVID-19 outbreak in 2020 and a reference period in 2019.**

|  | All consultations | Telephone consultations | Physical consultations | Home visits |
|---|---|---|---|---|
| 20-24/03/2020 | 235 | 148 | 69 | 18 |
| 27-30/03/2020 | 240 | 149 | 70 | 20 |
| 03-06/04/2020 | 192 | 107 | 66 | 19 |
| 10-13/04/2020 | 239 | 120 | 94 | 24 |
| 17-20/04/2020 | 129 | 66 | 52 | 12 |
| Average per weekend 2020 (all GPCs) | 222 | 118 | 70 | 19 |
| Average per weekend 2020 (GPCs with data in 2019 only) | 244 | 138 | 82 | 23 |
| Average per weekend 2019 reference group | 174 | 0* | 148 | 26 |

*: Before the pandemic, remote consultations were neither reimbursed nor deontologically allowed

p<0,001) and home visits (N = 51, 15%, p<0,001). For 969 (6%) of the patients follow-up by the own GP was recommended.

The average consultation rate was 222 per 100 000 citizens per weekend. The first two weekends had a similar workload but the third and fifth had a lower workload due to a decrease in the number of all types of consultations (see Table 2). Because it contains Eastern Monday, the small increase in the fourth weekend does not relate to an increased workload. Compared to 2019, the total amount of consultations in 2020 (data available for five GPCs) increased with 40% exclusively due to the arise of telephone consultations. The amount of physical consultations decreased by 45% and even more in the final weekend.

Among the patients who underwent a physical consultation, 2226 (42%) had a prior telephone consultation by a GP. Among the patients with a diagnose at risk for COVID-19 this was significantly more: 1027 (70%, p<0,001) had a prior telephone consultation. However,433 (30%) patients only spoke with a call taker. Before a home visit, 423 (30%) had a prior telephone consultation, again significantly more (N = 209, 62%, p<0,001) in case of a diagnosis suspected of COVID-19.

## Changes in the demand for care

See Table 3 for an overview of the changes in the rate of physical consultations by diagnosis (ICPC chapter). There was a small increase of consultations for psychological (including fear

**Table 3. Shift in diagnostic categories of patients in primary out of hours care during a COVID-19 outbreak (all ICPC-chapters with >50 cases).**

| ICPC-chapter | 2019 (n) | 2019 (per 100 000 citizens) | 2020 (n) | 2020 (per 100 000 citizens) | Change (%) |
|---|---|---|---|---|---|
| General | 720 | 83 | 402 | 46 | -44 |
| Digestive | 646 | 52 | 361 | 24 | -54 |
| Eye | 138 | 11 | 79 | 5 | -53 |
| Ear | 246 | 20 | 91 | 6 | -69 |
| Circulatory | 60 | 5 | 80 | 5 | 10 |
| Musculoskeletal | 401 | 32 | 313 | 21 | -36 |
| Neurological | 91 | 7 | 66 | 4 | -40 |
| Psychological | 56 | 4 | 82 | 5 | 21 |
| Respiratory | 1106 | 88 | 567 | 37 | -58 |
| Skin | 437 | 35 | 416 | 27 | -21 |
| Urology | 138 | 11 | 116 | 8 | -31 |

ICPC: International Catalogue of Primary Care

of COVID-19) and cardiovascular diagnoses whereas there was a marked decrease in the amount of physical consultations for respiratory, ophthalmological, digestive and ear-related diagnoses.

## Discussion

This study proves it is possible to rapidly collect reliable data about the characteristics of primary out-of-hours care. GPCs changed their way of working rapidly and profoundly almost immediately after the start of the Belgian lock-down. The participating GPCs made an impressing shift from no telephone consultations to half of all contacts being delivered by telephone. This organisational change is very similar to the yet unstudied response in other countries worldwide. [4–6]

At the start of the study half of the primary out-of-hours care was COVID-19 related. The GPs handled the vast majority of the patients themselves with a combination of telephone and physical consultations while referring 6% to secondary care. The demand for work certificates caused an additional increase in consultations.

Of the patients who had a diagnosis at risk for COVID-19 after a physical consultation, 30% did not have a prior telephone consultation (they only had a brief contact with a call taker). Given the fact that all GPCs state that within the infectious patient flow all patients get a GP telephone consultation first, most likely the majority of them have been seen within the regular flow. Possibly, a minority of these patients did get another type of COVID-19 risk assessment not registered in the software: prior consultation outside of the GPC (patient's own GP working in the weekend or specialist). This proportion of patients suspected of COVID-19 physically seen without prior telephone consultation is high and indicates a risk for the GPs working in the regular patient flow, as they do not wear personal protective equipment. Awaiting validated triage guidelines we recommend to give a telephone consultation to all patients and not to rely on call takers to differentiate between infectious and regular care.

During the study, the total workload decreased drastically: from a workload much higher as in 2019, entirely due to telephone consultations, to a total workload much lower than in 2019. One explanation is the reduced demand of information as citizens get more accustomed to the partial lock-down. Another is the reduced demand for medical certificates related to work. Finally it might be related to the outbreak itself as the number of hospitalisations followed a similar decline with a one week delay. These reasons however do not explain the halving of the demand for physical consultations. This is surely a direct or indirect effect of the outbreak as this demand has steadily gone up for many years. An indirect effect might be due to the lock-down: a decline in infections and accidents can be expected. The more pronounced decline of ophthalmological and digestive diagnoses might be due to the reduced amount of other infections caused by the lock-down.

Another indirect effect is the fear of patients to consult a medical service as described in Italy. [10] The decline in physical consultations is most pronounced for respiratory, ophthalmological, digestive and ear-related diagnoses probably also due to a combination of decreased infections and fear of consulting. The decline in respiratory diagnosis can be explained by the trend to handle these patients by telephone. This reduction in care unrelated to COVID-19 is an important finding because it might be correlated to increased morbidity and mortality.

The strength of this study lies in the large number of included consultations and the speed of reporting. This study has got several limitations: it only describes what happens but does not allow for any conclusions about the outcomes, efficacy or safety of these consultations. We have no data regarding the length of the (telephone) consultations or the deployment of additional staff. For telephone consultations, safety was already an issue before the COVID-19

pandemic. [21] During the study period, ambulatory testing patients for SARS-CoV-2 in primary care was not allowed in Belgium so the proportion of truly infected patients remains unknown. Rapid changes in the organisation of the GPCs reduced data quality: some consultations might not have been registered correctly and there might be small differences in organisation among the participating GPCs. We were unable to see which patients were in the infectious or the regular patient flow. The data fields for referral (to the own GP or the ED) are not compulsory to complete a report and thus prone to under registration.

This paper does not allow to make any predictions based upon this data. Because COVID-19 patients typically deteriorate one week after the first symptoms [22], the stress on the primary care system might be a predictor for overcrowding in the hospitals in the near future. Further study into the significance of the decline in regular care is needed as this might indicate a safety problem. When implementing new strategies to limit COVID-19, close attention should be given to its side effects on regular care.

The successful implementation of telephone consultations opens new possibilities for the post COVID-19 era but should be studied more profoundly, especially regarding patient's safety. We recommend longer follow-up studies within different healthcare systems and not restricted to out-of-hours care. The current study provides some arguments in favour of a primary care first model with remote consultations but more aspects such as financial consequences, total work burden, proportion of truly infected patients, morbidity and mortality should be studied.

## Conclusions

Belgian GPs have been able to rapidly re-organise care in order to handle a COVID-19 outbreak and provide a safe working environment. Initially half of the GP's consultations during out-of-hours care were related to COVID-19 leading to an increased work load followed by a workload below the normal average after five weeks. This provokes questions about patient's safety for care unrelated to COVID-19. GPs handled 40% of all out-of-hours consultations by telephone alone. Among the patients with a diagnosis at risk of COVID-19, 30% had a physical consultation without a prior telephone consultation to detect this risk. This implies a risk for unprotected staff providing care for possibly infected patients.

## Supporting information

**S1 Checklist. STROBE statement—checklist of items that should be included in reports of *cross-sectional studies*.**
(DOCX)

**S1 File.**
(PDF)

**S2 File.**
(PDF)

## Acknowledgments

The authors like to show their gratitude to all participating General Practice Cooperatives and their staff.

## Author Contributions

**Conceptualization:** Stefan Morreel, Hilde Philips, Veronique Verhoeven.

**Data curation:** Stefan Morreel.

**Formal analysis:** Stefan Morreel.

**Funding acquisition:** Stefan Morreel.

**Investigation:** Stefan Morreel, Hilde Philips, Veronique Verhoeven.

**Methodology:** Stefan Morreel, Hilde Philips, Veronique Verhoeven.

**Project administration:** Stefan Morreel, Hilde Philips, Veronique Verhoeven.

**Resources:** Stefan Morreel, Hilde Philips.

**Software:** Stefan Morreel.

**Supervision:** Hilde Philips, Veronique Verhoeven.

**Validation:** Stefan Morreel.

**Visualization:** Stefan Morreel.

**Writing – original draft:** Stefan Morreel.

**Writing – review & editing:** Stefan Morreel, Hilde Philips, Veronique Verhoeven.

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
