## [Decision Letter · Decision Letter 0]

15 Jun 2020

PONE-D-20-15141

Organisation and characteristics of out-of-hours primary care during a COVID-19 outbreak: a real-time observational study

PLOS ONE

Dear Dr. Morreel,

Thank you for submitting your manuscript to PLOS ONE. After careful consideration, we feel that it has merit but does not fully meet PLOS ONE’s publication criteria as it currently stands. Therefore, we invite you to submit a revised version of the manuscript that addresses the points raised during the review process.

We look forward to receiving your revised manuscript.

Kind regards,

Wen-Jun Tu

Academic Editor

PLOS ONE

Journal Requirements:

2. Thank you for including your competing interests statement; "I have read the journal's policy and the

authors of this manuscript have the following

competing interests:All authors have completed the ICMJE uniform disclosure form and declare: Stefan Morreel is an unpaid Board member of one of the participating General Practice Cooperatives and is paid by the Belgian ministry of help to coordinate the response of this cooperative to the COVID-19 pandemic. Veronique Verhoeven has participated at the same GPC by handling telephone calls in a fee-for-service model. All authors are board members of iCAREdata as part of their academic position, the database used in this study. "

Reviewers' comments:

Reviewer's Responses to Questions

**Comments to the Author**

1. Is the manuscript technically sound, and do the data support the conclusions?

Reviewer #1: Partly

Reviewer #2: Yes

2. Has the statistical analysis been performed appropriately and rigorously? 

Reviewer #1: Yes

Reviewer #2: Yes

3. Have the authors made all data underlying the findings in their manuscript fully available?

Reviewer #1: No

Reviewer #2: No

4. Is the manuscript presented in an intelligible fashion and written in standard English?

Reviewer #1: Yes

Reviewer #2: Yes

5. Review Comments to the Author

Reviewer #1: The study's strength lies in the massive data that allows health care system and policy makers to understand more about the nature of out-of-hours primary care during COVID19. However the data is limited in the sense that it only provides a descriptive analysis and lacking in in-depth discussion about how this data might have larger consequences to changing health care system in Belgium during pandemic, and compared with other countries' out-of-hours primary care. I suggest to expand the analysis to make connection to the larger/global implication.

Reviewer #2: The authors present an interesting case study on out-of-hours primary care in Belgium during the COVID-19 pandemic. Overall, the paper is well-written, but a proper copyediting should be performed before publication. I have a few remarks that authors should address in a revised version:

Page 5 (Sec Methods): how are the other 77% of the Flemish population covered? It seems to be quite a large number of GPs per centre, are they all on duty in shifts? Do they otherwise work full time as GPs and cover the same population?

Page 7 / Fig 1: the representation and explanation of the process during the pandemic should be clearer. All patients are advised to call first? What happens to those who just show up? It is stated on page 10, for example, that 30% did not have a telephone consultation. Please explain in more detail how you differentiate triage from consultation. How and when where patients informed about the new process?

Do you have time stamps / durations for the call handling and different consultation types? Did the durations differ in the two years? Are telephone consultations significantly faster than physical consultations? What was the staff utilisation in the two periods?

Page 8 (Sec Characteristics, first paragraph): What were the remaining 3%?

Page 10, Table 2: Why weren’t there any telephone consultations last year? Was it not possible to call the centre before the pandemic? Was everyone seen who called the centre?

Page 10, Table 2: could you provide some numbers on the pandemic for the 5 weeks you studied to get a better idea about the situation in Belgium during that time? From the news we got the impression that Belgium was hit comparatively hard by the virus.

Page 10, last sentence: do you have any explanation why ear-related diagnoses are on the list of those with the highest decline? From my experience, I would have imagined something like back pain or headaches etc. to be on that list.

From your findings, would you conclude that the centres should keep telephone consultations as an option even after COVID-19? Can you make conclusions based on your findings and your data?

Can you make any remarks on other European countries that also provide out-of-hours care by GPs like Germany? Did you find any papers or comments on that? Would you say that the care process should be transferable to other countries and would you suggest to use it?

I am missing an outlook at the end of the paper. What should follow-up research study? How can the results be used in future research?

6. PLOS authors have the option to publish the peer review history of their article (what does this mean?). If published, this will include your full peer review and any attached files.

Reviewer #1: No

Reviewer #2: No

---

## [Author Response · Author response to Decision Letter 0]

25 Jun 2020

We thank the reviewers for their generous comments on the manuscript and have edited the manuscript accordingly. For your convenience we have made a table with the suggestions of the reviewers and our adaptations/comments in the rebuttal letter. 

Change competing interests statement We have added the suggestion to the cover letter. For clarity we have also copied our data availability statement to the cover letter.

Lacking in in-depth discussion about how this data might have larger consequences to changing health care system in Belgium during pandemic This suggestion is similar to more specific critics of the second reviewer so we have commented those

Page 5 (Sec Methods): how are the other 77% of the Flemish population covered? It seems to be quite a large number of GPs per centre, are they all on duty in shifts? Do they otherwise work full time as GPs and cover the same population? We have added the requested information. We have clarified that the studied GPCs cover 23% of the Flemish population. The remaining 77% is covered by GPCs we could not include or have the old system of rota groups for on call duties. We believe adding information about the large amount of included GPs would make the methodology section to long but provide it here for your interest: Belgium has got a rather large amount of GPs per capita and the included GPCs have a lot of part-time working GPs and trainees because of the proximity of two academic departments. GPs with a chronic illness or above 65 years of age are not obliged to work in the GPCs, all the others are.

Page 7 / Fig 1: the representation and explanation of the process during the pandemic should be clearer. All patients are advised to call first? We have entirely rewritten this paragraph and simplified Fig 1. It is now more clear that all patients need to call (but some will only speak with a call taker). 

It is stated on page 10, for example, that 30% did not have a telephone consultation. We clarified this statement: 433 (30%) only spoke with a call taker

Please explain in more detail how you differentiate triage from consultation. New sentence: “In two GPCs, the call takers refer selected patients directly to the ED (using a local triage protocol) while all other GPCs do not perform any kind of triage.”

How and when where patients informed about the new process? Added to the introduction: The government’s recommendation to always call a doctor prior to going to a practice or ED was omnipresent in the media.

Do you have time stamps / durations for the call handling and different consultation types? Did the durations differ in the two years? Are telephone consultations significantly faster than physical consultations? We do not have this data, added to limitations.

What was the staff utilisation in the two periods? The GPCs definitely increased their staff (GPs and call takers/receptionists) but we do not have data proving this, neither was it part of our questionnaire. We have added this to the limitations.

Page 8 (Sec Characteristics, first paragraph): What were the remaining 3%? There is not remaining 3%, we have clarified this sentence.

Page 10, Table 2: Why weren’t there any telephone consultations last year? Was it not possible to call the centre before the pandemic? Was everyone seen who called the centre? Added to the introduction and to the footnote of this table: “Before the pandemic, remote consultations were neither reimbursed nor deontologically allowed.”

Page 10, Table 2: could you provide some numbers on the pandemic for the 5 weeks you studied to get a better idea about the situation in Belgium during that time? From the news we got the impression that Belgium was hit comparatively hard by the virus. We have extended the epidemiological information in the introduction. Belgium surely was hit hard but we have chosen not to compare these data to other countries or regions because Belgium is one of the few countries including many suspected cases to the total number of cases. Currently there are no peer reviewed publications available about the comparison of the burden of this pandemic to individual countries or regions. It would be more correct to compare Belgium to New-York or Northern Italy then to compare the numbers to bigger countries.

Page 10, last sentence: do you have any explanation why ear-related diagnoses are on the list of those with the highest decline? From my experience, I would have imagined something like back pain or headaches etc. to be on that list. There was a decline for back pain (included in the ICPC-chapter Musculoskeletal) and headache (chapter Neurological) but similar to the overall decline so we do not discuss them separately.

Added to the discussion: “The more pronounced decline of ophthalmological and digestive diagnoses is probably due to the reduced amount of other infections caused by the lock-down.”

From your findings, would you conclude that the centres should keep telephone consultations as an option even after COVID-19? Can you make conclusions based on your findings and your data? Our data do not allow for definitive conclusions but are an interesting starting point. Added to discussion: “The successful implementation of telephone consultations opens new possibilities for the post COVID-19 era but should be studied more profoundly, especially regarding patient’s safety.”

Can you make any remarks on other European countries that also provide out-of-hours care by GPs like Germany? Did you find any papers or comments on that? Would you say that the care process should be transferable to other countries and would you suggest to use it? We have entirely rewritten the first paragraph of the Discussion with additional references. This study alone does not allow for a recommendation on the transfer of this care process to other countries although is surely brings in one argument.

I am missing an outlook at the end of the paper. What should follow-up research study? How can the results be used in future research? Such an outlook has been added to the final paragraph of the discussion.

We hope that the manuscript is now suitable for publication in Plos One.

Sincerely,

The authors

---

## [Decision Letter · Decision Letter 1]

20 Jul 2020

PONE-D-20-15141R1

Organisation and characteristics of out-of-hours primary care during a COVID-19 outbreak: a real-time observational study

PLOS ONE

Dear Dr. Morreel,

Thank you for submitting your manuscript to PLOS ONE. After careful consideration, we feel that it has merit but does not fully meet PLOS ONE’s publication criteria as it currently stands. Therefore, we invite you to submit a revised version of the manuscript that addresses the points raised during the review process.

We look forward to receiving your revised manuscript.

Kind regards,

Wen-Jun Tu

Academic Editor

PLOS ONE

Additional Editor Comments (if provided):

1. In order to provide a more complete information to our readers on the topic, we would like to emphasize the importance to cross referencing very recent material on the same topic published in "PLoS ONE ". Therefore, it would be highly appreciated if you would check the contents published in the last two years of "PLoS ONE" (https://journals.plos.org/plosone/) and add all material relevant to your article to the reference list.

2. add “Clinical features and short-term outcomes of 102 patients with corona virus disease 2019 in Wuhan, China. Clinical Infectious Diseases.” in the revision text

Reviewers' comments:

Reviewer's Responses to Questions

**Comments to the Author**

1. If the authors have adequately addressed your comments raised in a previous round of review and you feel that this manuscript is now acceptable for publication, you may indicate that here to bypass the “Comments to the Author” section, enter your conflict of interest statement in the “Confidential to Editor” section, and submit your "Accept" recommendation.

Reviewer #2: (No Response)

2. Is the manuscript technically sound, and do the data support the conclusions?

Reviewer #2: Yes

3. Has the statistical analysis been performed appropriately and rigorously? 

Reviewer #2: Yes

4. Have the authors made all data underlying the findings in their manuscript fully available?

Reviewer #2: No

5. Is the manuscript presented in an intelligible fashion and written in standard English?

Reviewer #2: (No Response)

6. Review Comments to the Author

Reviewer #2: I think the authors did a good job revising the manuscript. I just have a few minor comments:

- The information and references about other countries that were added to the discussion might be better moved to the intro section as part of the relevant literature.

- The last sentence of the added outlook should be revised.

- The paper needs final proofreading.

7. PLOS authors have the option to publish the peer review history of their article (what does this mean?). If published, this will include your full peer review and any attached files.

Reviewer #2: No

---

## [Author Response · Author response to Decision Letter 1]

27 Jul 2020

Dear editor,

We thank the reviewers for their additional comments on the manuscript and have edited the manuscript accordingly. We believe this paper is now ready for publication and will contribute to the wide diversity of knowledge about COVID-19 distributed by PLoS ONE as it is the first study within the COVID-19 collection with a primary care perspective. For your convenience we have made a table with the suggestions of the reviewers and our adaptations/comments.

Comment Our adaptation and/or comment

In order to provide a more complete information to our readers on the topic, we would like to emphasize the importance to cross referencing very recent material on the same topic published in "PLoS ONE ". We have added a recent PLoS ONE study to the Methodology section (Grant MC, Geoghegan L, Arbyn M, et al. The prevalence of symptoms in 24,410 adults infected by the novel coronavirus (SARS-CoV-2; COVID-19): A systematic review and meta-analysis of 148 studies from 9 countries. PLoS One. 2020;15(6):e0234765. Published 2020 Jun 23. doi:10.1371/journal.pone.0234765)

Editor: add “Clinical features and short-term outcomes of 102 patients with corona virus disease 2019 in Wuhan, China. Clinical Infectious Diseases.” in the revision text We have added this reference to the introduction

The information and references about other countries that were added to the discussion might be better moved to the intro section as part of the relevant literature. We have moved this paragraph as requested

The last sentence of the added outlook should be revised. This sentence was indeed incomplete and has been corrected

The paper needs final proofreading. We have proofread the entire article and made some improvements here and there

We hope that the manuscript is now suitable for publication in Plos One.

Sincerely,

The authors

---

## [Editor Report · Decision Letter 2]

31 Jul 2020

Organisation and characteristics of out-of-hours primary care during a COVID-19 outbreak: a real-time observational study

PONE-D-20-15141R2

Dear Dr. Morreel,

We’re pleased to inform you that your manuscript has been judged scientifically suitable for publication and will be formally accepted for publication once it meets all outstanding technical requirements.

Kind regards,

Wen-Jun Tu

Academic Editor

PLOS ONE
---

## [Editor Report · Acceptance letter]

4 Aug 2020

PONE-D-20-15141R2 

Organisation and characteristics of out-of-hours primary care during a COVID-19 outbreak: a real-time observational study 

Dear Dr. Morreel:

I'm pleased to inform you that your manuscript has been deemed suitable for publication in PLOS ONE. Congratulations! Your manuscript is now with our production department. 

Kind regards, 

on behalf of

Dr. Wen-Jun Tu 

Academic Editor

PLOS ONE